# ZGUL: Zero-shot Generalization to Unseen Languages using Multi-source Ensembling of Language Adapters

**Vipul Rathore    Rajdeep Dhingra    Parag Singla    Mausam**
Indian Institute of Technology
New Delhi, India
`rathorevipul28@gmail.com, rajdeep.cse.iitd@gmail.com`
`parags@cse.iitd.ac.in, mausam@cse.iitd.ac.in`

## Abstract

We tackle the problem of zero-shot cross-lingual transfer in NLP tasks via the use of language adapters (LAs). Most of the earlier works have explored training with adapter of a single source (often English), and testing either using the target LA or LA of another related language. Training target LA requires unlabeled data, which may not be readily available for low resource *unseen* languages: those that are neither seen by the underlying multilingual language model (e.g., mBERT), nor do we have any (labeled or unlabeled) data for them.

We posit that for more effective cross-lingual transfer, instead of just one source LA, we need to leverage LAs of multiple (linguistically or geographically related) source languages, both at train and test-time – which we investigate via our novel neural architecture, ZGUL. Extensive experimentation across four language groups, covering 15 unseen target languages, demonstrates improvements of up to 3.2 average F1 points over standard fine-tuning and other strong baselines on POS tagging and NER tasks. We also extend ZGUL to settings where either (1) some unlabeled data or (2) few-shot training examples are available for the target language. We find that ZGUL continues to outperform baselines in these settings too.

## 1 Introduction

Massive multilingual pretrained language models (PLMs) such as mBERT (Devlin et al., 2019) and XLM-R (Conneau et al., 2020) support 100+ languages. We are motivated by the vision of extending NLP to the thousands (Muller et al., 2021) of *unseen* languages, i.e., those not present in PLMs, and for which unlabeled corpus is also not readily available. A natural approach is *zero-shot cross-lingual transfer* – train the model on one or more source languages and test on the target language, zero-shot. Two common approaches are (1) *Standard Fine Tuning* (SFT) - fine-tune all parameters

of a PLM on task-specific training data in source language(s), and (2) *Language Adapters* (LAs) – small trainable modules inserted within a PLM transformer, and trained on target language's unlabeled data using Masked Language Modeling (MLM). At test time, in SFT the fine-tuned PLM is applied directly to target language inputs, whereas for the latter, the LA of source language is replaced with that of target language for better zero-shot performance. Unfortunately, while in the former case, one would expect presence of unlabeled data for a target language to pre-train the PLM for good performance, the latter also requires the same unlabeled data for training a target adapter.

For the large number of low resource languages, curating a decent-sized unlabeled corpus is a challenge. For instance, there are only 291 languages that have Wikipedias with over 1,000 articles. Consequently, existing works use English data for training on the task and use the English LA (Pfeiffer et al., 2020b; He et al., 2021) or an ensemble of related language LAs (Wang et al., 2021) at inference time. We posit that this is sub-optimal; for better performance, we should leverage multiple source languages (ideally, related to target language) and their LAs, *both* at train and test-time.

To this end, we propose ZGUL, **Z**ero-shot **G**eneralization to **U**nseen **L**anguages, which explores this hypothesis.[1] It has three main components. First, it fuses LAs from source languages at train time by leveraging *AdapterFusion* (Pfeiffer et al., 2021a), which was originally developed for fusing multiple task adapters. This allows ZGUL to *locally* decide the relevance of each LA for each token in each layer. Second, ZGUL leverages the typological properties of languages (encoded in existing language vectors) as addi-

---

[1]Following previous work (Wang et al., 2021), we assume the target language has the same script as one or more languages in the PLM, so that the model does not have to deal with a sequence of [UNK] tokens at test time. We present the script statistics for our target languages in App. Table 14.

tional information for computing *global* LA attention scores. Finally, ZGUL also implements the Entropy-Minimization (EM)-based test-time tuning of LA attention weights (Wang et al., 2021).

We denote a language *group* as a set of phylogenetically or demographically close languages, similar to Wang et al. (2021). We experiment on 15 unseen languages from four language groups: Slavic, Germanic, African and Indo-Aryan, on POS tagging and NER tasks. In each group, we train on multiple (3 to 4) source languages (including English), for which task-specific training data and LAs are available. ZGUL obtains substantial improvements on unseen languages compared to strong baselines like SFT and CPG (Conditional Parameter Generation (Üstün et al., 2020)), in a purely zero-shot setting. Detailed ablations show the importance of each component in ZGUL. We perform attention analysis to assess if learned weights to source languages' LAs are consistent with their *relatedness* to the target language.

Further, we study two additional scenarios, where (1) some unlabeled data, and (2) some task-specific training data are available for the target language. We extend ZGUL in these settings and find that our extensions continue to outperform our competitive baselines, including ones that use unlabeled data for a target language to either (1) pre-train mBERT or (2) train target's LA.

Our contributions can be summarized as: (1) We propose a strong method (ZGUL) to combine the pretrained language adapters during training itself. To the best of our knowledge, we are the first to systematically attempt this in context of LAs. ZGUL further incorporates test-time tuning of LA weights using Entropy Minimization (EM). (2) ZGUL outperforms the competitive multi-source baselines for zero-shot transfer on languages unseen in mBERT. (3) ZGUL exhibits a strong correlation between learned attention scores to adapters and the linguistic relatedness between source and target languages. (4) ZGUL achieves competitive results in a few-shot setting, where a limited amount of labeled (target language) training data is available. (5) When target language unlabeled data is available, a modification of ZGUL outperforms baselines for nine out of twelve languages. To encourage reproducibility, we publicly release our code and trained models.[2]

## 2 Related Work

**Single-source Adapter Tuning:** We build on MAD-X (Pfeiffer et al., 2020b), which introduces two phases of adapter training. 1. Pretraining Language adapter (LA) for each language $L_i$: inserting an LA in each layer of transformer model $\mathcal{M}$ (denoted by $\mathcal{L}_i \circ \mathcal{M}$) and training on unlabeled data for language $\mathcal{L}_i$ using the MLM objective. 2. Training TA for a task $T_j$: stacking LA for source language $L_{src}$ with TA for task $T_j$ (denoted by $\mathcal{T}_j \circ \mathcal{L}_{src} \circ \mathcal{M}$), in which $\mathcal{T}_j$ and the task-specific prediction head are the only trainable parameters. During inference, $\mathcal{L}_{src}$ is replaced with $\mathcal{L}_{tgt}$, i.e. $\mathcal{T}_j \circ \mathcal{L}_{tgt} \circ \mathcal{M}$ is used. The MAD-X paradigm uses only one LA for a given input sentence. Also, it assumes the availability of $\mathcal{L}_{tgt}$. If not available, English adapter (He et al., 2021; Pfeiffer et al., 2020b) or a related language's adapter (Wang et al., 2021) is used at test-time.

**Adapter Combination:** Pfeiffer et al. (2021a) introduce *AdapterFusion*, a technique that combines multiple pretrained TAs $\mathcal{T}_1, ... \mathcal{T}_n$ to solve a new target task $T_{n+1}$. It learns the attention weights of $\mathcal{T}_1, ... \mathcal{T}_n$ while being fine-tuned on the data for $T_{n+1}$. Vu et al. (2022) adapt this technique for fusing domains and testing on out-of-domain data. This technique has not been applied in the context of LAs so far. The recent release of 50 LAs on AdapterHub[3] enables studying this for LAs.

Recently, Wang et al. (2021) propose EMEA (Entropy Minimized Ensembling of Adapters) for efficiently combining multiple LAs at inference time. EMEA calculates the entropy of the prediction during test time and adjusts the LA attention scores (initialized uniformly) using Gradient Descent, aiming to give higher importance to the LA that increases the confidence score of the prediction. However, the training is still conducted using English as a single source.

**Generation of LA using Shared Parameters:** Üstün et al. (2020) employ the Conditional Parameter Generation (CPG) technique (Platanios et al., 2018) for training on multiple source languages. They utilize a CPG module, referred to as *CPGAdapter*, which takes a typological language vector as input and generates a Language Adapter (LA). The CPGAdapter is shared across all source languages and trained from scratch for a specific task. Since an LA is determined by the input lan-

---

guage's vector, this approach can directly generalize to unseen languages. However, it is worth noting that CPG is data-intensive as it learns the parameters of the CPGAdapter from scratch.

We note that CPG comes under a broader category of hypernetworks that generate weights for a larger main network (Ha et al., 2016), which have been recently explored successfully for task mixing (Karimi Mahabadi et al., 2021). In our experiments, we include a comparison with the CPG method.

## 3 Model for Ensembling of Adapters

Our goal is to combine a set of source LAs for optimal zero-shot cross-lingual transfer to unseen languages, which are neither in mBERT, nor have readily available labeled or unlabeled data. Similar to previous works (Pfeiffer et al., 2020b; Wang et al., 2021), we focus on languages whose scripts are seen in mBERT. Handling unseen languages with unseen scripts is a more challenging task, which we leave for future research.

Our approach can be described using two high-level components: (1) train-time ensembling and (2) test-time ensembling, explained below for every layer $l$ (for notational simplicity, we skip using $l$ in notations but clarify wherever required).

### 3.1 Train-time Ensembling

During training, we make use of an attention mechanism inspired by the combination of *task* adapters explored in Pfeiffer et al. (2021a). While they focus on creating an ensemble of task adapters, our focus is on combining *language* adapters. Additionally, we identify valuable information available in typological language vectors (Littell et al., 2017), that we can leverage. To achieve this, we design two sub-components in our architecture, which we later combine for each layer (refer to Figure 1).

**Token-based Attention (FUSION):** This sub-network computes the *local* attention weights over source LAs for each token using the output of the feed forward layer as its query, and the individual language adapters' outputs as both the key and the value. Mathematically, for $t^{th}$ token, the embedding obtained after passing through feed forward layer becomes the query vector $q^{(t)}$. The individual LAs' outputs following this become key (and value) matrices $K^{(t)}$ (and $V^{(t)}$). The attention weights of source LAs for $t^{th}$ token are computed using the dot-product attention between projected query $W_q q^{(t)}$ and projected key matrix $W_k K^{(t)}$:

$$\alpha_F^{(t)} = Softmax((W_q q^{(t)})^T (W_k K^{(t)}))$$

The FUSION output for $t^{th}$ token is given by:

$$o_F^{(t)} = \alpha_F^{(t)} \odot (W_v V^{(t)})$$

Here, $W_q, W_k, W_v$ are projection matrices which are different for every layer $l$.

**Language-vector-based Attention (LANG2VEC):** This sub-network computes the *global* attention weights for source LAs using the input language vector (of the token) as the query, the language vectors of the source languages as the keys, and the outputs through individual LAs as the values. Mathematically, the attention weights over the LAs are obtained through a projected dot-product attention between the input language vector $l_{inp}$ as query vector and the source language vectors stacked as a key matrix $L_{src}$.

Here the language vectors are derived by passing the language features $lf$[4] (each feature being binary) through a single-layer trainable MLP:

$$l_{inp} = \text{MLP}(lf_{inp})$$

The LANG2VEC attention scores for $t^{th}$ token are given by:

$$\alpha_L^{(t)} = Softmax((l_{lang[t]})^T W_L (L_{src}))$$

Here $W_L$ is a projection matrix associated differently with each layer $l$. $lang[t]$ denotes language vector of $t^{th}$ token in the input

The output of $t^{th}$ token is given by:

$$o_L^{(t)} = \alpha_L^{(t)} \odot (W_v V^{(t)})$$

Since, $lang[t]$ is same across all tokens in an example and also across all examples in a language, we refer LANG2VEC attention as *global*. On the other hand, FUSION computes *local* attention scores that depend purely on the token-level outputs of the feed forward layer and hence are local to each token in any given input sentence.

**Combining the two ensembling modules:** We pass the input sentence through both networks, and for $t^{th}$ token receive the outputs $o_F^{(t)}$ and $o_L^{(t)}$, corresponding to the FUSION and LANG2VEC networks, respectively. These two vectors are concatenated and passed through a fully connected layer. The output of this linear layer, denoted as $o_{LA}^{(t)}$, serves as input to the task adapter (TA) to obtain the final output $o_{final}^{(t)}$.

$$o_{LA}^{(t)} = LinearLayer(o_F^{(t)} \oplus o_L^{(t)})$$
$$o_{final}^{(t)} = TA(o_{LA}^{(t)})$$

The above process is repeated for each layer $l$ in

---

[4]Following the implementation of Üstün et al. (2020), we use 103-dimensional 'syntax'-based features available at https://github.com/antonisa/lang2vec.

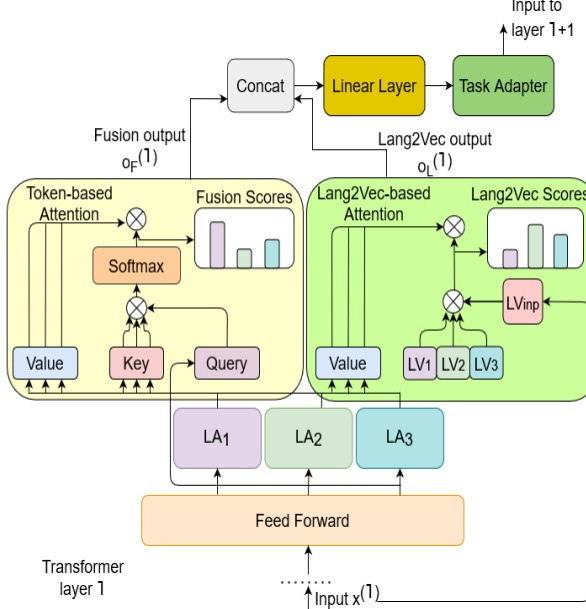

Figure 1: FUSION Network (left) and LANG2VEC Network (right) outputs are concatenated and sent to a Linear layer followed by a TA in every transformer layer $l$

the transformer architecture. We note that the LAs are kept frozen throughout the training process, while only the TA and other parameters described in FUSION and LANG2VEC modules being trainable. The training objective is word-level cross-entropy loss for all models. For model selection, we evaluate ZGUL and other baseline models on the combined dev set of source languages' data. We do not use any dev set of target languages, as doing so would violate the zero-shot assumption on target language data.

### 3.2 Test-time Ensembling

Wang et al. (2021) introduced EMEA, an inference-time Entropy Minimization (EM)-based algorithm to adjust the LA attention scores, initializing them from uniform (as mentioned in sec. 2). In our case, since we learn the attention scores during training itself, we seek to further leverage the EM algorithm by initializing them with our learnt networks' weights. First, we compute the entropy of ZGUL's predicted labels, averaged over all words in the input sentence, using an initial forward pass of our trained model. Since ZGUL has two different attention-based networks – FUSION and LANG2VEC, it's trainable parameters are the attention weights for both these networks. We back-propagate the computed entropy and update both these attention weights using SGD optimization. In the next iteration, entropy is computed again using

a forward pass with the modified attention weights. This process is repeated for $T$ iterations, where $T$ and learning rate $lr$ are the hyperparameters, which are tuned on dev set of linguistically most related (based on distributed similarity, shown in figure 3) source language for each target (grid search details in Appendix A). Detailed EMEA algorithm is presented in Algo. 1 (Appendix).

## 4 Experiments

We aim to address the following questions. (1) How does ZGUL perform in a zero-shot setting compared to the other baselines on unseen languages? What is the incremental contribution of ZGUL's components to the performance on LRLs? (2) Are LA attention weights learnt by ZGUL interpretable, i.e., whether genetically/syntactically more similar source languages get higher attention scores? (3) How does ZGUL's performance change after incorporating unlabelled target language data? (4) How does ZGUL's performance vary in a *few-shot* setting, where a few training examples of the target language are provided to the model for fine-tuning?

### 4.1 Datasets, Tasks and Baselines

**Datasets and Tasks:** We experiment with 4 diverse language groups: Germanic, Slavic, African and Indo-Aryan. Following previous works (Wang et al., 2021; Pfeiffer et al., 2020b), we choose named entity recognition (NER) and part-of-speech (POS) tagging tasks. We select target languages which are unseen in mBERT subject to their availability of test sets for each task. This leads us to a total of 15 target languages spanning Germanic and Slavic for POS, and African and Indo-Aryan for NER. For African and Indo-Aryan NER experiments, we use the MasakhaNER (Adelani et al., 2021) and WikiAnn (Pan et al., 2017) datasets respectively. For POS experiments, we use Universal Treebank 2.5 (Nivre et al., 2020). We pick training languages from each group that have pre-trained adapters available. The details of training and test languages, as well as corresponding task for each group are presented in Table 1. For detailed statistics, please refer to tables 13, 14, 16.

**Baselines:** We experiment with two sets of baselines. In the first set, the baselines use only English as single source language during training:

- **SFT-En**: Standard mBERT fine-tuning on English

| Language Group | Train set* | Test set | Task |
|---|---|---|---|
| Germanic | German(De), Islandic(Is) | Faroese(Fo), Gothic(Got), Swiss German(Gsw) | POS |
| Slavic | Russian(Ru), Czech(Cs) | Pomak(Qpm), Upper Sorbian(Hsb) Old East Slavic(Orv), Old Church Slavonic(Cu) | POS |
| African | Amharic(Amh) Swahili(Swa), Wolof(Wol) | Hausa(Hau), Igbo(Ibo), Kinyarwanda(Kin) Luganda(Lug), Luo(Luo), Nigerian Pidgin(Pcm) | NER |
| Indo-Aryan | Hindi(Hi), Bengali(Bn), Urdu(Ur) | Assamese(As), Bhojpuri(Bh) | NER |

Table 1: Language groups or sets of related source and target languages, along with tasks. *English (En) is added to train set in each case.

- **MADX-{$\mathcal{S}$}**: Training with En LA (with trainable TA stacked on top) and using strategy $\mathcal{S}$ during inference, where $\mathcal{S}$ can be one of: (1) **En**: En LA at inference, (2) **Rel**: Using linguistically most related (based on similarity in Table 3) LA from source languages and (3) **EMEA**[5]: Ensembling all source LAs and perform EM (initialized with uniform)

In the second set, we compare against models trained upon multiple source languages (belonging to a group), in addition to English:

- **SFT-M**: Standard fine-tuning on data from all source languages.

- **CPG**: Conditional Parameter Generation (Üstün et al., 2020), see Section 2.

- **MADX$^{multi}$-{$\mathcal{S}$}**: We naturally extend MADX-En to the multi-source scenario by dynamically switching on the LA corresponding to the input sentence's language during training. $\mathcal{S}$ refers to inference strategy that can be one of the following (as described in English baselines above): En, Rel, Uniform Ensembling or EMEA ensembling.

It is important to note that the EM algorithm is not applicable to SFT and CPG baselines because SFT does not use an adapter, while CPG has only a single (shared) adapter. Consequently, there are no ensemble weights that can be tuned during inference for these methods. The EM algorithm is a distinctive feature of the ensemble-based methods like ZGUL, which allows for further optimization and performance improvement.

**Evaluation Metric:** We report micro-F1 evaluated on each token using seqeval toolkit (Nakayama, 2018). For all experiments, we report the average F1 from three training runs of the models with three different random seeds. The standard deviation is reported in Appendix G.

### 4.2 Results: Zero-Shot Transfer

Tables 2 and 3 present experimental findings for Germanic and Slavic POS, as well as African and Indo-Aryan NER, respectively. ZGUL outperforms other baselines for 10 out of 15 unseen test languages. In terms of POS, ZGUL achieves a respectable gain of 1.8 average F1 points for Germanic and a marginal improvement of 0.4 points for Slavic compared to its closest baseline CPG – the gains being particularly impressive for Gothic, Swiss German and Pomak languages. For NER, ZGUL achieves decent gains of 3.2 points and 0.9 points for the Indo-Aryan and the African groups respectively over the closest baseline i.e. SFT-M – the gains being upto 4 F1 points for Luo. Moreover, baselines trained on a single En source perform significantly worse (upto 24 points gap in Indo-Aryan), highlighting the importance of multi-source training for effective cross-lingual transfer. We note that CPG outperfoms SFT-M for POS tagging, but order switches for NER. This is to be expected due to huge number of parameters in CPG (details in Sec. A) and the smaller sizes of NER datasets compared to POS (details in App. 13).

We also observe a substantial performance gap between MADX$^{multi}$-Rel and ZGUL – the former performing upto 7.5 points average F1 worse than ZGUL for the Indo-Aryan group. This demonstrates that relying solely on the most related LA is sub-optimal compared to ZGUL, which leverages aggregated information from multiple LAs. Additionally, MADX$^{multi}$-Uniform, which does a naive averaging of LAs, performs even worse overall. Though MADX$^{multi}$-EMEA shows some improvement over it, yet remains below ZGUL's performance by an average of about 4 points over all languages. This finding highlights the effectiveness of ZGUL-style training, as the EM algorithm benefits from an informed initialization of weights, rather than a naive uniform initialization strategy.

---

[5]We do same grid search for EMEA as ZGUL. Ref. to A

|  | **Germanic** | | | | | **Slavic** | | | |
|  | Fo | Got | Gsw | **Avg** | | Qpm | Hsb | Orv | Cu | **Avg** |
|---|---|---|---|---|---|---|---|---|---|---|
| SFT-En | 72.4 | 13.2 | 55.1 | 46.9 | | 40 | 64.8 | 56.5 | 28 | 47.3 |
| MADX-En | 71.6 | 15.6 | 57.1 | 48.1 | | 41.8 | 63.2 | 54.3 | 29.5 | 47.2 |
| MADX-Rel | 72.8 | 15.2 | 57 | 48.3 | | 39.3 | 60.2 | 57 | 32.2 | 47.2 |
| MADX-EMEA | 74 | 14.8 | 55.3 | 48 | | 42.1 | 63.1 | 56.4 | 32 | 48.4 |
| MADX$^{multi}$-Rel | 75.8 | 14.7 | 60.4 | 50.3 | | 47.2 | 74.8 | 62.9 | **35.8** | 55.2 |
| MADX$^{multi}$-Uniform | 75.6 | 8.3 | 56.4 | 46.8 | | 46.4 | 74.6 | 62.1 | 34.3 | 54.4 |
| MADX$^{multi}$-EMEA | 76.2 | 12.5 | 62.1 | 50.3 | | 48 | 75 | 63.2 | 34.7 | 55.2 |
| SFT-M | **77.3** | 16.8 | 61.8 | 52.0 | | 47.6 | 75.4 | 63.7 | 34.7 | 55.4 |
| CPG | 77 | 16 | 63.6 | 52.2 | | 46.9 | 76.7 | **64.1** | 35.6 | 55.8 |
| ZGUL | 76.9 | **20.2** | **64.8** | **54$^*$** | | **50.7** | **76.8** | 63 | 34.4 | **56.2** |
| ZGUL w/o EM | 76.8 | 14.9 | 63 | 51.6 | | 49.7 | 76 | 62.6 | 33.7 | 55.5 |
| ZGUL w/o L | 76.9 | 17.8 | 61.3 | 52 | | 49.8 | 76 | 62.8 | 33.3 | 55.5 |
| ZGUL w/o F | 76.9 | 18.7 | 62.4 | 52.7 | | 49.8 | 76.5 | 63 | 33.7 | 55.8 |

Table 2: F1 of POS Tagging Results for Germanic and Slavic language groups, $^*$ denotes p-value < 0.005 for McNemar's test on aggregated results over all test languages in a group

|  | **African** | | | | | | | | **Indo-Aryan** | | |
|  | Hau | Ibo | Kin | Lug | Luo | Pcm | **Avg** | | As | Bh | **Avg** |
|---|---|---|---|---|---|---|---|---|---|---|---|
| SFT-En | 43.2 | 47.1 | 49.1 | 47.5 | 29 | 64.4 | 46.7 | | 32.1 | 35.7 | 33.9 |
| MADX-En | 41.3 | 51.2 | 46 | 49.8 | 29.2 | 64.5 | 47 | | 33 | 44.8 | 38.9 |
| MADX-Rel | 39.1 | 49.1 | 46.3 | 48.7 | 29.7 | 65.1 | 46.3 | | 42.4 | 46.9 | 44.7 |
| MADX-EMEA | 42 | 52.9 | 50.2 | 50.6 | 30.7 | 65.5 | 48.7 | | 41.6 | 47.6 | 44.6 |
| MADX$^{multi}$-Rel | 47.8 | 49.6 | 49.6 | 46.8 | 31.8 | 62.4 | 48 | | 61.7 | 61.9 | 61.8 |
| MADX$^{multi}$-Uniform | 47 | 51.1 | 50.8 | 49.9 | 31.7 | 62.7 | 48.7 | | 60 | 59 | 59.5 |
| MADX$^{multi}$-EMEA | 47.4 | 54.4 | 55 | 51.1 | 32.8 | 63.9 | 50.8 | | 62.4 | 59.7 | 61.1 |
| SFT-M | 51.3 | 55.7 | **57.9** | **56.0** | 36.0 | 65.0 | 53.7 | | 70.8 | 61.4 | 66.1 |
| CPG | 49 | 51.1 | 55.1 | 54 | 34.2 | 65.7 | 51.5 | | 62 | 63.3 | 62.7 |
| ZGUL | **53.6** | **56.8** | 56.2 | 54.2 | **40.2** | **66.5** | **54.6$^*$** | | **74.4** | **64.1** | **69.3$^*$** |
| ZGUL w/o EM | 52.4 | 55.6 | 57.3 | 55.1 | 38.6 | 65.5 | 54.1 | | 71.1 | 58 | 64.6 |
| ZGUL w/o L | 49 | 54.7 | 57.3 | 53.5 | 35.5 | 65.5 | 52.6 | | 61.5 | 64.2 | 62.9 |
| ZGUL w/o F | 53.1 | 54.1 | 57.7 | 55.7 | 36.6 | 65.6 | 53.8 | | 67.2 | 61.7 | 64.5 |

Table 3: F1 of NER Results for African and Indo-Aryan language groups, $^*$ denotes p-value < 0.005 for McNemar's test on aggregated results over all test languages in a group

More analysis on this follows in Sec. 4.3.

Ablation results in last three rows in Tables 2, 3 examine the impact of each of the 3 components, FUSION, LANG2VEC and Entropy Minimization (EM), on ZGUL's performance. We observe a positive impact of each component for each language group, in terms of average F1 scores. For individual languages as well, we see an improvement in F1 due to each component, exceptions being Kinyarwanda and Luganda, where EM marginally hurts the performance. This could occur when wrong predictions are confident ones, and further performing EM over those predictions might hurt the overall performance.

### 4.3 Interpretability w.r.t. Attention Scores

We wanted to examine if the attention weights computed by ZGUL are interpretable. In order to do

this, we computed the correlation[6] between the (final) attention scores computed by ZGUL at inference time, and the language relatedness with the source, for each of the test languages. Since ZGUL has two different networks for computing the attention scores, i.e., the Fusion network and LANG2VEC, we correspondingly compute the correlation with respect to the average attention scores in both these networks. For both networks, we compute the average of scores across all tokens, layers as well as examples in a target language. To compute the language relatedness, we use the distributed similarity metric obtained as the average of the syntactic and genetic similarities (we refer to Appendix C for details). Table 4 presents the results. Clearly, we observe a high correlation between the attention scores computed by ZGUL

---

[6]Pearson Product-Moment Correlation

| Group | LANG2VEC | FUSION | $M^m$-EM |
|-------|----------|--------|----------|
| Germanic | 0.57 | 0.45 | 0.23 |
| Slavic | 0.93 | 0.85 | 0.14 |
| African | 0.61 | 0.2 | 0.18 |
| Indo-Aryan | 0.81 | 0.67 | -0.01 |

Table 4: Correlation between ZGUL's attention scores and syntactic-genetic (averaged) similarity of source-target pairs for both LANG2VEC and FUSION networks and for the EMEA (multi-source) baseline

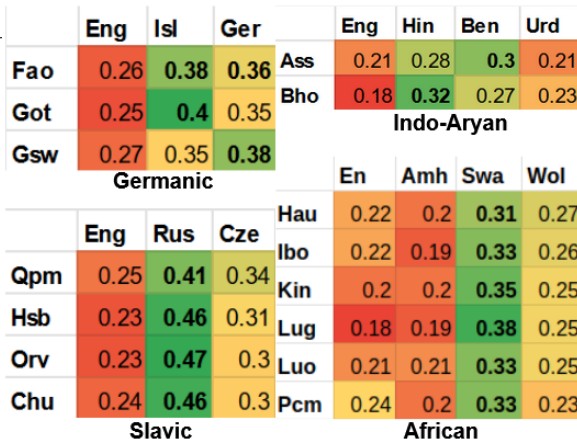

Figure 2: LANG2VEC attention scores for each of the test languages (clustered group-wise).

Figure 3: Language relatedness between target and source languages in each group, computed as average of syntactic and genetic similarity metrics.

and language relatedness, especially for Slavic and Indo-Aryan groups, for both the attention networks. This means that the model is assigning higher attention score to languages which are more related in a linguistic sense, or in other words, language relatedness can be thought of as a reasonable proxy, for deciding how much (relative) importance to give to LA from each of the source languages for building an efficient predictive model (for our tasks). Further, we note that among the two networks, the scores are particularly higher for LANG2VEC, which we attribute to the fact that the network explicitly uses language features as its input, and therefore is in a better position to directly capture language relatedness, compared to the Fusion network, which has to rely on this information implicitly through the tokens in each language.

For the sake of comparison, we also include the correlations for MADX$^{multi}$-EMEA model (referred to as M$^m$-EM for brevity), which does LA ensembling purely at inference time, to contrast the impact of learning ensemble weights via training in ZGUL on correlation with language relatedness. Clearly, the scores are significantly lower in this case, pointing to the fact that EMEA alone is not able to capture a good notion of language relatedness, which possibly explains its weaker performance as well (as observed in Tables 2 & 3).

For completeness, Figures 2 & 3 present the attention scores for the LANG2VEC network, and also the language relatedness to the source languages, for each test language, grouped by corresponding language group. The similarity in the two heat maps (depicted via color coding) again corroborates the high correlation between the attention scores computed by ZGUL with language relatedness.

### 4.4 Leveraging Unlabeled Target Data

Is ZGUL useful in case some amount of unlabeled data is available for the target language? To answer this, we use Wikipedia dumps, which are available for 12 out of our 15 target languages. For each language $L^{tgt}$, (1) we train its Language Adapter $LA^{tgt}$, and (2) pre-train mBERT model using MLM, denoted as $mBERT^{tgt}$. We also extend ZGUL to ZGUL++ as follows: we initialize ZGUL's encoder weights with $mBERT^{tgt}$ and fine-tune it with the additional adapter $LA^{tgt}$, inserted along with other source LAs. This is trained only on source languages' task-specific data (as target language training is not available). We compare ZGUL++ with (1) MADX$^{multi}$-Tgt, which trains MADX in multi-source fashion and at inference, use the $LA^{tgt}$, and (2) SFT++, which initializes SFT's encoder weights with $mBERT^{tgt}$ and fine-tunes on the source languages' data.

Table 5 shows ZGUL++'s average gains of 2 F1 points over our competitive baseline SFT++. ZGUL++ achieves SOTA performance for 9 of

| | Fo | Got | Gsw | Hsb | Cu | Hau | Ibo | Kin | Lug | Pcm | As | Bh | **Avg** |
|---|---|---|---|---|---|---|---|---|---|---|---|---|---|
| MADX$^{multi}$-Tgt | 81.8 | 15.5 | 76.1 | 87.6 | 32.8 | 69.7 | 64 | 53.1 | 54.8 | 61.8 | 57.7 | 64.4 | 59.9 |
| SFT++ | 81.9 | 16.5 | 76.1 | **91.5** | 34.6 | 75.8 | **71.8** | 68 | 64 | 66.9 | 68.1 | 65.9 | 65.1 |
| ZGUL++ | **82.3** | **17.9** | **82.7** | **91.5** | **35.4** | **77.7** | 71.4 | 69.2 | **64.4** | **67.6** | 73.5 | **70.5** | **67** |
| Z.++ w/o Tgt. LA | 82.1 | 17.8 | 79.7 | 90.5 | 35.1 | 77.1 | 71.7 | **69.7** | 63.7 | 67.2 | **74.1** | 67.9 | 66.4 |
| Z.++ w/o mBERT$^{tgt}$ | 78.5 | 15.8 | 69.5 | 87.1 | 34.2 | 64.1 | 60.4 | 60.5 | 56.2 | 65.8 | 64.6 | 66.7 | 60.3 |
| Unlabeled datasize | 160k | 2k | 100k | 150k | 8k | 350k | 230k | 30k | 35k | 8k | 250k | 45k | |

Table 5: F1 scores after incorporating target unlabeled data in various models. Unlabeled datasize is in # sentences.

12 languages while it's ablated variant (not using $LA^{tgt}$) does so for two more languages. The gains for Swiss German (6.6 F1 points) are particularly impressive. Ablations for ZGUL++ show that though incorporating the $LA^{tgt}$ is beneficial with average gain of about 0.6 F1 point over all languages, the crucial component is initializing with $mBERT^{tgt}$, which leads to around 7 avg. F1 point gains. Hence, the additional target pre-training step is crucial, in conjunction with utilizing the target LA, for effectively exploiting the unlabeled data. We investigate how the performance scales for each model from zero-shot setting (no unlabeled data) to utilizing 100% of the Wikipedia target data. We sample 2 bins, containing 25% and 50% of the full target data. We then plot the average F1 scores over all 12 languages for each bin in fig. 4.

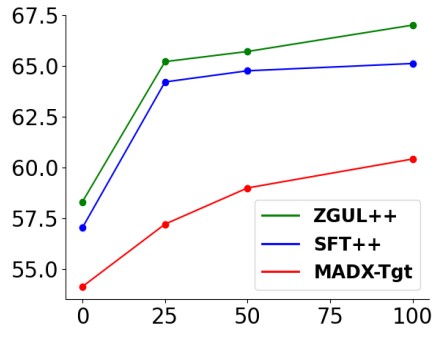

Figure 4: Average F1 scores over target languages w.r.t. percentage of Wikipedia data used. 0 denotes zero-shot.

We observe ZGUL++ is effective across the regime. Compared to SFT++, the gains are higher on the 100% regime, while compared to MADX-Tgt baseline, a steep gain is observed upon just using 25% data.

### 4.5 Few-Shot Performance

In this experiment, we take the trained ZGUL and other multi-source models, i.e., CPG, SFT-M, and fine-tune them further for a few labeled examples from the train set of each target language. We do this for those test languages, whose training set is available (12 out of 15). We sample training bins of sizes 10, 30, 70 and 100 samples. We observe

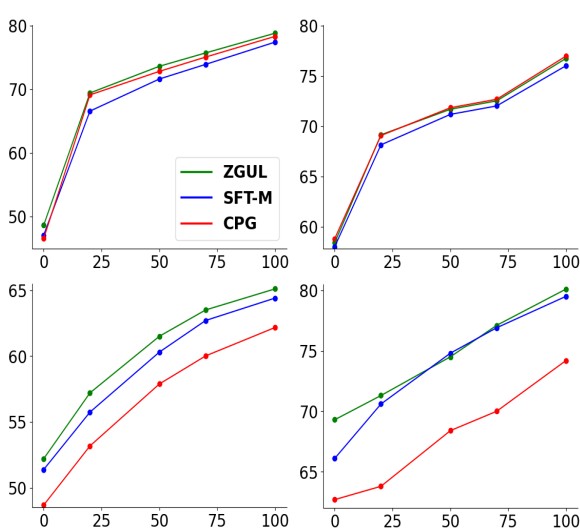

Figure 5: Few-shot F1 averaged over languages in a group for various few-shot bins. Top row: Germanic, Slavic. Bottom row: African, Indo-Aryan.

in Fig. 5 that ZGUL scales smoothly for all language groups, maintaining its dominance over the baselines in each case, except for Slavic, where its performance is similar to CPG baseline. The relative ordering of baselines, i.e. CPG outperforming SFT-M for Slavic and Germanic (POS), and SFT-M outperforming CPG for African and Indo-Aryan (NER), is also maintained across the regime of few-shot samples, similar to zero-shot setting. The learning plateau is not reached in the curves for either of the language groups, showing that adding more target examples would likely result in further improvement of all the models, albeit at a smaller pace. We present the detailed language-wise few-shot curves in Appendix H.

### 4.6 EM tuning using Target Dev set

In the purely zero-shot setting, we tuned EM parameters for ensemble-based methods, i.e. MADX$^{multi}$-EMEA and ZGUL, on the closest source language's dev set. However, if we assume

|  | **Germanic** | | | | | **Slavic** | | | | |
|  | Fo | Got | Gsw | **Avg** | | Qpm | Hsb | Orv | Cu | **Avg** |
|---|---|---|---|---|---|---|---|---|---|---|
| Closest baseline (CPG) | 77 | 16 | 63.6 | 52.2 | ‖ | 46.9 | 76.7 | **64.1** | 35.6 | 55.8 |
| MADX$^{multi}$-EMEA | **77.1** | 14 | 62.4 | 51.2 | ‖ | 49.7 | 75.4 | 63.4 | **35.7** | 56.1 |
| ZGUL | **77.1** | **22.6** | **65.1** | **54.9** | ‖ | **51.3** | **77.4** | 63.2 | 34.7 | **56.7** |

Table 6: POS Tagging Results for Germanic and Slavic groups when utilizing target dev set for EM tuning

|  | **African** | | | | | | | **Indo-Aryan** | | |
|  | Hau | Ibo | Kin | Lug | Luo | Pcm | **Avg** | As | Bh | **Avg** |
|---|---|---|---|---|---|---|---|---|---|---|
| Closest baseline (SFT-M) | 51.3 | 55.7 | **57.9** | **56.0** | 36.0 | 65.0 | 53.7 | 70.8 | 61.4 | 66.1 |
| MADX$^{multi}$-EMEA | 49.3 | 55.2 | 55 | 52.4 | 34.2 | 66.3 | 52.1 | 63.6 | 62.8 | 63.2 |
| ZGUL | **54.3** | **57.5** | 56.9 | 54.8 | **40.9** | **66.5** | **55.2** | **76.9** | **64.5** | **70.7** |

Table 7: NER Results for African and Indo-Aryan groups when utilizing target dev set for EM tuning

the target dev set availability, which indeed holds for our target languages, one can leverage it for EM hyperparameter tuning. We present the results for the same in tables 6 and 7. The gains of ZGUL become more pronounced, obtaining up to 1.4 avg. F1 points in the Indo-Aryan group.

## 5 Conclusion and Future Work

We present ZGUL[7], a novel neural model for ensembling the pre-trained language adapters (LAs) for multi-source training. This is performed by fusing the LAs at train-time to compute *local* token-level attention scores, along with typological language vectors to compute a second *global* attention score, which are combined for effective training. Entropy Minimization (EM) is carried out at test-time to further refine those attention scores. Our model obtains strong performance for languages unseen by mBERT but with the seen scripts. We present various analyses including that the learnt attention weights have significant correlation with linguistic similarity between source and target, and demonstrating scalability of our model in the unlabeled data and few-shot labeled data settings as well.

In the future, our approach, being task-agnostic, can be applied to more non-trivial tasks, such as generation (Kolluru et al., 2022, 2021), semantic parsing (Awasthi et al., 2023), relation extraction (Rathore et al., 2022; Bhartiya et al., 2022), and knowledge graph completion (Chakrabarti et al., 2022; Mittal et al., 2023). Our technique may complement other approaches for morphologically rich languages (Nzeyimana and Rubungo, 2022) and for

---
[7]https://github.com/dair-iitd/ZGUL

those with scripts unseen in mBERT (Pfeiffer et al., 2021b). Extending our approach to code-switched data, in which each input token can potentially belong to a different language, is another interesting future direction.

## Acknowledgements

Mausam is supported by grants from Google, Huawei, and Jai Gupta Chair Professorship from IIT Delhi. Parag was supported by the DARPA Explainable Artificial Intelligence (XAI) Program (#N66001-17-2-4032). Mausam and Parag are also supported by IBM SUR awards. Vipul was supported by Prime Minister's Research Fellowship (PMRF). We thank IIT Delhi HPC facility for compute resources. We thank Keshav Kolluru for helpful comments on earlier drafts of the paper. Any opinions, findings, conclusions or recommendations expressed here are those of the authors and do not necessarily reflect the views or official policies, either expressed or implied, of the funding agencies.

## Limitations

Our method incurs high inference-time overhead for each forward pass owing to Adapters being inserted in each layer. Further, the entropy minimization typically needs 5 or 10 forward passes for effective performance, which leads to further multiplying factor with each forward pass time. These trade-offs between performance and cost are inherited from (Wang et al., 2021) itself. Language Adapters have been trained on the Wikipedia dump of source/target languages. This might potentially impose restrictions to extending our technique's

efficacy to domain-specific tasks not having suffi-
cient publicly available data in that language and
domain as to train a strong Adapter (E.g. Medical
domain for African languages). Presently, our tech-
nique cannot be tested directly on unseen scripts
because our tokenization/embedding layer is same
as that of mBERT and may become a bottleneck for
Adapters to directly perform well. Our approach
is not currently tested on deep semantic tasks and
generation-based tasks owing to the lack of suitable
large-scale datasets for evaluation.

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

Methods in Natural Language Processing, EMNLP
2022, Abu Dhabi, United Arab Emirates, Decem-
ber 7-11, 2022*, pages 11922–11938. Association for
Computational Linguistics.

Alexis Conneau, Kartikay Khandelwal, Naman Goyal,
Vishrav Chaudhary, Guillaume Wenzek, Francisco
Guzmán, Édouard Grave, Myle Ott, Luke Zettle-
moyer, and Veselin Stoyanov. 2020. Unsupervised
cross-lingual representation learning at scale. In *Pro-
ceedings of the 58th Annual Meeting of the Asso-
ciation for Computational Linguistics*, pages 8440–
8451.

Jacob Devlin, Ming-Wei Chang, Kenton Lee, and
Kristina Toutanova. 2019. Bert: Pre-training of deep
bidirectional transformers for language understand-
ing. In *Proceedings of the 2019 Conference of the

North American Chapter of the Association for Com-
putational Linguistics: Human Language Technolo-
gies, Volume 1 (Long and Short Papers)*, pages 4171–
4186.

David Ha, Andrew Dai, and Quoc V Le. 2016. Hyper-
networks. *arXiv preprint arXiv:1609.09106*.

Ruidan He, Linlin Liu, Hai Ye, Qingyu Tan, Bosheng
Ding, Liying Cheng, Jiawei Low, Lidong Bing, and
Luo Si. 2021. On the effectiveness of adapter-based
tuning for pretrained language model adaptation. In
*Proceedings of the 59th Annual Meeting of the Asso-
ciation for Computational Linguistics and the 11th
International Joint Conference on Natural Language
Processing (Volume 1: Long Papers)*, pages 2208–
2222.

Rabeeh Karimi Mahabadi, Sebastian Ruder, Mostafa
Dehghani, and James Henderson. 2021. Parameter-
efficient multi-task fine-tuning for transformers via
shared hypernetworks. In *Proceedings of the 59th
Annual Meeting of the Association for Computational
Linguistics and the 11th International Joint Confer-
ence on Natural Language Processing (Volume 1:
Long Papers)*, pages 565–576, Online. Association
for Computational Linguistics.

Keshav Kolluru, Muqeeth Mohammed, Shubham
Mittal, Soumen Chakrabarti, and Mausam. 2022.
Alignment-augmented consistent translation for mul-
tilingual open information extraction. In *Proceedings
of the 60th Annual Meeting of the Association for
Computational Linguistics (Volume 1: Long Papers)*,
pages 2502–2517, Dublin, Ireland. Association for
Computational Linguistics.

Keshav Kolluru, Martin Rezk, Pat Verga, William W
Cohen, and Partha Talukdar. 2021. Multilingual fact
linking. In *3rd Conference on Automated Knowledge
Base Construction*.

Patrick Littell, David R. Mortensen, Ke Lin, Katherine
Kairis, Carlisle Turner, and Lori Levin. 2017. URIEL
and lang2vec: Representing languages as typological,
geographical, and phylogenetic vectors. In *Proceed-
ings of the 15th Conference of the European Chap-
ter of the Association for Computational Linguistics:
Volume 2, Short Papers*, pages 8–14, Valencia, Spain.
Association for Computational Linguistics.

Shubham Mittal, Keshav Kolluru, Soumen Chakrabarti,
and Mausam. 2023. mokb6: A multilingual open
knowledge base completion benchmark. In *Proceed-
ings of the 61st Annual Meeting of the Association for
Computational Linguistics (Volume 2: Short Papers),
ACL 2023, Toronto, Canada, July 9-14, 2023*, pages
201–214. Association for Computational Linguistics.

Benjamin Muller, Antonios Anastasopoulos, Benoît
Sagot, and Djamé Seddah. 2021. When being un-
seen from mbert is just the beginning: Handling new
languages with multilingual language models. In
*Proceedings of the 2021 Conference of the North

*American Chapter of the Association for Computational Linguistics: Human Language Technologies*, pages 448–462.

Hiroki Nakayama. 2018. seqeval: A python framework for sequence labeling evaluation. *Software available from https://github. com/chakki-works/seqeval*.

Joakim Nivre, Marie-Catherine de Marneffe, Filip Ginter, Jan Hajic, Christopher D Manning, Sampo Pyysalo, Sebastian Schuster, Francis Tyers, and Daniel Zeman. 2020. Universal dependencies v2: An evergrowing multilingual treebank collection. In *Proceedings of the 12th Language Resources and Evaluation Conference*, pages 4034–4043.

Antoine Nzeyimana and Andre Niyongabo Rubungo. 2022. Kinyabert: a morphology-aware kinyarwanda language model. In *Proceedings of the 60th Annual Meeting of the Association for Computational Linguistics (Volume 1: Long Papers)*, pages 5347–5363.

Xiaoman Pan, Boliang Zhang, Jonathan May, Joel Nothman, Kevin Knight, and Heng Ji. 2017. Cross-lingual name tagging and linking for 282 languages. In *Proceedings of the 55th Annual Meeting of the Association for Computational Linguistics (Volume 1: Long Papers)*, pages 1946–1958.

Jonas Pfeiffer, Aishwarya Kamath, Andreas Rücklé, Kyunghyun Cho, and Iryna Gurevych. 2021a. Adapterfusion: Non-destructive task composition for transfer learning. In *Proceedings of the 16th Conference of the European Chapter of the Association for Computational Linguistics: Main Volume*, pages 487–503.

Jonas Pfeiffer, Andreas Rücklé, Clifton Poth, Aishwarya Kamath, Ivan Vulić, Sebastian Ruder, Kyunghyun Cho, and Iryna Gurevych. 2020a. AdapterHub: A framework for adapting transformers. In *Proceedings of the 2020 Conference on Empirical Methods in Natural Language Processing: System Demonstrations*, Online. Association for Computational Linguistics.

Jonas Pfeiffer, Ivan Vulić, Iryna Gurevych, and Sebastian Ruder. 2020b. MAD-X: An Adapter-Based Framework for Multi-Task Cross-Lingual Transfer. In *Proceedings of the 2020 Conference on Empirical Methods in Natural Language Processing (EMNLP)*, pages 7654–7673, Online. Association for Computational Linguistics.

Jonas Pfeiffer, Ivan Vulić, Iryna Gurevych, and Sebastian Ruder. 2021b. Unks everywhere: Adapting multilingual language models to new scripts. In *Proceedings of the 2021 Conference on Empirical Methods in Natural Language Processing*, pages 10186–10203.

Emmanouil Antonios Platanios, Mrinmaya Sachan, Graham Neubig, and Tom Mitchell. 2018. Contextual parameter generation for universal neural machine translation. In *Proceedings of the 2018 Conference on Empirical Methods in Natural Language Processing*, pages 425–435, Brussels, Belgium. Association for Computational Linguistics.

Vipul Rathore, Kartikeya Badola, Parag Singla, and Mausam . 2022. PARE: A simple and strong baseline for monolingual and multilingual distantly supervised relation extraction. In *Proceedings of the 60th Annual Meeting of the Association for Computational Linguistics (Volume 2: Short Papers)*, pages 340–354, Dublin, Ireland. Association for Computational Linguistics.

Ahmet Üstün, Arianna Bisazza, Gosse Bouma, and Gertjan van Noord. 2020. UDapter: Language adaptation for truly Universal Dependency parsing. In *Proceedings of the 2020 Conference on Empirical Methods in Natural Language Processing (EMNLP)*, pages 2302–2315, Online. Association for Computational Linguistics.

Thuy-Trang Vu, Shahram Khadivi, Dinh Phung, and Gholamreza Haffari. 2022. Domain generalisation of NMT: Fusing adapters with leave-one-domain-out training. In *Findings of the Association for Computational Linguistics: ACL 2022*, pages 582–588.

Xinyi Wang, Yulia Tsvetkov, Sebastian Ruder, and Graham Neubig. 2021. Efficient test time adapter ensembling for low-resource language varieties. In *Findings of the Association for Computational Linguistics: EMNLP 2021*, pages 730–737.

# A  Hyperparameter and Implementation Details

We use a Tesla V100 GPU (32 GB) for training all models. We use AdapterHub [8] (Pfeiffer et al., 2020a) for all our experiments and analysis. We mention the hyperparameter grid for SFT and all adapter-based models (including ZGUL) in table 8. For all experiments, we report the average of 3 training runs of the models (with 3 different random seeds). We tune the Adapter Reduction Factor (RF) in the range of {3,4} but generally find 3 to be the best for all adapter-based methods on all datasets. The EM algorithm used in both ZGUL and EMEA uses grid search in the range {1, 5, 10} for iterations $T$ and {0.05, 0.1, 0.5, 1.0} for learning rate $lr$.

| Hyperparameter | SFT | Adapter Based |
|---|---|---|
| Learning Rate | {2e-5, 3e-5, 5e-5} | {5e-5, 1e-4} |
| Max. epochs | 10 for NER, 5 for POS | 10 for NER, 5 for POS |
| Reduction Factor | NIL | {3, 4} |
| Batch Size | {16, 32} | {16, 32} |
| EM Steps | NIL | {1, 5, 10} |
| EM LR | NIL | {0.05, 0.1, 0.5, 1.0} |
| Few-shot LR | {1e-5, 5e-5, 1e-4} | {1e-5, 5e-5, 1e-4} |
| Few-shot epochs | {1, 5, 10} | {1, 5, 10} |
| Few-shot Batch Size | {1, 4, 8} | {1, 4, 8} |

Table 8: Hyperparameter grids for ours models

## A.1  Trainable Parameters & Training time

| Model | Params | Per-Epoch Training time (in mins) | | | | Avg |
|---|---|---|---|---|---|---|
| | | Indo-Aryan | African | Germanic | Slavic | |
| SFT-M | 177M | 9 | 3 | 36 | 30 | 19.5 |
| CPG | 253M | 37 | 13 | 141 | 115 | 76.5 |
| ZGUL | 41M | 14 | 5 | 36 | 29 | 21 |

Table 9: Trainable Parameters & per-epoch training time of all models

## A.2  Detailed EM Algorithm used in ZGUL

---
**Algorithm 1** EM algorithm during Inference

---
**Input:** Our model's scores $\alpha^0 = (\alpha_F^0, \alpha_L^0)$, test data $x$, learning rate $lr$, update steps $T$
**Output:** Prediction $\hat{y}$
1: **function** EMEA++()
2:     **for** $t \leftarrow 0$ to $T-1$ **do**
3:         $\beta^t \leftarrow Softmax(\alpha^t)$            ▷ Normalize the LA scores
4:         $H(x, \alpha) \leftarrow Entropy(TA \circ L_{wavg}(h, \beta^t) \circ M)$     ▷ Compute Entropy
5:         $g_F^t = \nabla_\alpha H(x, \alpha_F^t)$     ▷ Compute Token Attention gradient
6:         $\alpha_F^{t+1} \leftarrow Update(\alpha_F^t, g_F^t)$     ▷ Update Token Attention weights
7:         $g_L^t = \nabla_\alpha H(x, \alpha_L^t)$     ▷ Compute LangVec Attention gradient
8:         $\alpha_L^{t+1} \leftarrow Update(\alpha_L^t, g_L^t)$     ▷ Update LangVec Attention weights
9:         $\alpha^{t+1} = (\alpha_F^{t+1}, \alpha_L^{t+1})$
10:     **end for**
11:     $\alpha^T \leftarrow Softmax(\alpha^T)$     ▷ Final Normalize
12:     $\hat{y} \leftarrow Predict(TA \circ L_{wavg}(h, \alpha^T) \circ M))$     ▷ Compute Prediction
13: **end function**

---

We note that we made the following amendments to the originaly proposed EMEA (Wang et al., 2021) for our setting - (1) we made each token-level attention weights in each layer trainable, while the original EMEA had tied it layer-wise. This gives the EM method more degree of freedom in our framework compared to EMEA. (2) we have 2 attention networks, each initialized with its respective trained attention

---
[8] https://github.com/adapter-hub

| | Hau | Ibo | Kin | Lug | Luo | Pcm | As | Bh | Fo | Got | Gsw | Qpm | Hsb | Orv | Cu | **Avg.** |
|---|---|---|---|---|---|---|---|---|---|---|---|---|---|---|---|---|
| Layer-wise Tied (original EMEA) | 53.2 | 56.9 | 55.2 | 54.1 | 39.8 | 65.8 | 74.1 | 64.1 | 76.4 | 18.6 | 61.9 | 49.5 | 76.7 | 62.3 | 33.4 | 56.1 |
| ZGUL | 53.6 | 56.8 | 56.2 | 54.2 | 40.2 | 66.5 | 74.4 | 64.1 | 76.9 | 20.2 | 64.8 | 50.1 | 77.2 | 63 | 34.1 | 56.8 |

Table 10: Ablation showing why keeping token-wise trainable weights in EMEA is marginally better than tying it layer-wise

scores, while the original technique had only a single network, initialized naively with uniform weights. We found amendment (1) to be marginally better than the one when we tie the token-level weights layer-wise (as in EMEA), as shown in table 10. Also, amendment (2) naturally follows from the fact that our architecture has 2 components for LA ensembling.

## B  Robustness with XLM-Roberta

We wanted to examine whether our findings related to ZGUL's superior performance over the existing baselines carry over to other language models. Specifically, we trained ZGUL and other multi-source baselines on XLM-R Base model available on Adapterhub (Pfeiffer et al., 2020a) for Germanic and Indo-Aryan language groups. We had all the adapters for languages in these two groups, except for Bengali (Indo-Aryan) which we eliminated during training. Table 11 presents our findings. ZGUL beats both the baselines on both the language groups, with a gain of 1.6 pts on the Indo-Aryan group, and a gain of 0.9 pts on the Germanic group, compared to its closest competitor. Ablation analysis conforms to the trend observed with mBERT with EM being the most important component followed by LANG2VEC and FUSION being the least important. This clearly confirms the finding that ZGUL's gains are not particularly restricted to a specific choice of PLM.

| Model | Indo-Aryan | | | Germanic | | | |
|---|---|---|---|---|---|---|---|
| | As | Bh | **Avg** | Fo | Got | Gsw | **Avg** |
| SFT-M | **60.4** | 60.5 | 60.5 | 78 | 15.4 | 54.9 | 49.4 |
| CPG | 55.7 | 60.3 | 58 | **78.1** | 18.3 | **57.6** | 51.3 |
| ZGUL | 59.3 | **64.9** | **62.1** | 77.8 | **21.7** | 57.1 | **52.2** |
| ZG−EM | 58.1 | 61.5 | 59.8 | 77.6 | 15.3 | 54.9 | 49.3 |
| ZG−F | 60.3 | 62.1 | 61.2 | 77.6 | 21 | 54.7 | 51.1 |
| ZG−L | 59.7 | 58.7 | 59.2 | 77.8 | 21.1 | 54.2 | 51 |

Table 11: F1 of NER Results for Indo-Aryan (trained on En,Hi,Ur) and POS for Germanic models (trained on En, Is, De) with XLM-R-Base Adapters. − means without

## C  Quantifying similarity between source and target languages

We use the phlogenetic and syntactic distances between source and target languages for reference.[9]

### C.1  Target language assignment

We justify assigning the unseen target language to a group using nearest neighbour based on phylogenetic similarity (shown in fig. 6). E.g. Hausa is genetically most similar to Amharic, so it's been assigned to the African group. On the other hand, Luo has equal genetic similarity with all source languages, so we refer to the syntactic similarity (fig. 7), for tie-break, in which it's most similar to Wolof (also English, in this case, which is common in every group), and hence it's been assigned to the African group.

### C.2  Consistency with the learnt attention scores

We also make use of these similarity metrics to validate the attention scores being learnt in the Lang2Vec modules (explained in detail in section 4.3). E.g. for Bhojpuri, highest attention score goes to Hindi

---

[9]https://github.com/antonisa/lang2vec

Figure 6: Heatmap - Genetic similarity

| Target/S | Eng | Amh | Swa | Wol | Rus | Cze | Isl | Ger | Ara | Hin | Ben |
|---|---|---|---|---|---|---|---|---|---|---|---|
| Bho | 0.1 | 0 | 0 | 0 | 0.2 | 0.14 | 0.14 | 0.14 | 0 | 0.37 | 0.37 |
| Hau | 0 | 0.12 | 0 | 0 | 0 | 0 | 0 | 0 | 0 | 0 | 0 |
| Ibo | 0 | 0 | 0.5 | 0.17 | 0 | 0 | 0 | 0 | 0 | 0 | 0 |
| Kin | 0 | 0 | 0.62 | 0.2 | 0 | 0 | 0 | 0 | 0 | 0 | 0 |
| Lug | 0 | 0 | 0.62 | 0.2 | 0 | 0 | 0 | 0 | 0 | 0 | 0 |
| Luo | 0 | 0 | 0 | 0 | 0 | 0 | 0 | 0 | 0 | 0 | 0 |
| Pcm | 0.9 | 0 | 0 | 0 | 0.2 | 0.14 | 0.43 | 0.57 | 0.08 | 0.12 | 0.12 |
| Ass | 0.1 | 0 | 0 | 0 | 0.2 | 0.14 | 0.14 | 0.14 | 0 | 0.37 | 0.75 |
| Qpm | 0.14 | 0 | 0 | 0 | 0.43 | 0.43 | 0.14 | 0.14 | 0 | 0.14 | 0.12 |
| Hsb | 0.1 | 0 | 0 | 0 | 0.6 | 0.57 | 0.14 | 0.14 | 0 | 0.12 | 0.12 |
| Orv | 0.1 | 0 | 0 | 0 | 0.6 | 0.43 | 0.14 | 0.14 | 0 | 0.12 | 0.12 |
| Chu | 0.1 | 0 | 0 | 0 | 0.6 | 0.43 | 0.14 | 0.14 | 0 | 0.12 | 0.12 |
| Fao | 0.3 | 0 | 0 | 0 | 0.2 | 0.14 | 0.86 | 0.43 | 0 | 0.12 | 0.12 |
| Got | 0.2 | 0 | 0 | 0 | 0.2 | 0.14 | 0.29 | 0.29 | 0 | 0.12 | 0.12 |
| Gsw | 0.4 | 0 | 0 | 0 | 0.2 | 0.14 | 0.43 | 0.57 | 0 | 0.12 | 0.12 |

Figure 7: Heatmap - Syntactic similarity

| Target/S | Eng | Amh | Swa | Wol | Rus | Cze | Isl | Ger | Urd | Hin | Ben |
|---|---|---|---|---|---|---|---|---|---|---|---|
| Bho | 0.32 | 0.37 | 0.16 | 0.22 | 0.39 | 0.37 | 0.32 | 0.38 | 0.69 | 0.56 | 0.45 |
| Hau | 0.55 | 0.35 | 0.44 | 0.55 | 0.41 | 0.28 | 0.49 | 0.46 | 0.22 | 0.38 | 0.22 |
| Ibo | 0.39 | 0.24 | 0.33 | 0.44 | 0.28 | 0.29 | 0.39 | 0.32 | 0.16 | 0.17 | 0.18 |
| Kin | 0.23 | 0.21 | 0.34 | 0.23 | 0.21 | 0.29 | 0.28 | 0.19 | 0.14 | 0.13 | 0.16 |
| Lug | 0.22 | 0.22 | 0.52 | 0.3 | 0.22 | 0.33 | 0.32 | 0.23 | 0.09 | 0.13 | 0.2 |
| Luo | 0.37 | 0.34 | 0.46 | 0.51 | 0.31 | 0.33 | 0.41 | 0.34 | 0.23 | 0.26 | 0.17 |
| Pcm | 0.43 | 0.43 | 0.4 | 0.38 | 0.43 | 0.34 | 0.4 | 0.42 | 0.31 | 0.42 | 0.33 |
| Ass | 0.19 | 0.28 | 0.2 | 0.13 | 0.24 | 0.31 | 0.26 | 0.3 | 0.33 | 0.33 | 0.38 |
| Qpm | 0.52 | 0.38 | 0.32 | 0.46 | 0.54 | 0.45 | 0.41 | 0.46 | 0.42 | 0.41 | 0.31 |
| Hsb | 0.43 | 0.43 | 0.4 | 0.38 | 0.43 | 0.34 | 0.4 | 0.42 | 0.31 | 0.42 | 0.33 |
| Orv | 0.43 | 0.43 | 0.4 | 0.38 | 0.43 | 0.34 | 0.4 | 0.42 | 0.31 | 0.42 | 0.33 |
| Chu | 0.43 | 0.43 | 0.4 | 0.38 | 0.43 | 0.34 | 0.4 | 0.42 | 0.31 | 0.42 | 0.33 |
| Fao | 0.39 | 0.3 | 0.26 | 0.38 | 0.36 | 0.46 | 0.43 | 0.41 | 0.26 | 0.25 | 0.31 |
| Got | 0.33 | 0.29 | 0.33 | 0.38 | 0.33 | 0.39 | 0.42 | 0.4 | 0.29 | 0.29 | 0.3 |
| Gsw | 0.37 | 0.34 | 0.24 | 0.35 | 0.37 | 0.46 | 0.33 | 0.44 | 0.4 | 0.35 | 0.37 |

Figure 8: Various similarity metrics between source and target languages (higher the more similar). This is used to validate the assignment of the target languages to corresponding groups as well as for depicting correlation with the LA attention scores learnt by LANG2VEC component.

LA while for Assamese, it does for Bengali. Indeed, we observe that Bhojpuri is closest to Hindi while Assamese being closest to Bengali, based upon the average of both genetic and syntactic similarities (fig. 3). Detailed correlation between the attention scores and similarity have been discussed in section 4.3

## D  Datasets' details & Stats

| Code | Language |
|------|----------|
| Amh | Amharic |
| Ar | Arabic |
| As | Assamese |
| Be | Belorussian |
| Bg | Bulgarian |
| Bh | Bhojpuri |
| Bn | Bengali |
| Cs | Czech |
| Cu | Old Church Slavonic |
| Da | Danish |
| De | German |
| En | English |
| Fo | Faroese |
| Got | Gothic |
| Gsw | Swiss German |
| Hau | Hausa |
| Hi | Hindi |

| Code | Language |
|------|----------|
| Hsb | Upper Sorbian |
| Ibo | Igbo |
| Is | Icelandic |
| Kin | Kinyarwanda |
| Lug | Ganda |
| Luo | Luo |
| Mr | Marathi |
| No | Norwegian |
| Orv | Old East Slavik |
| Qpm | Pomak |
| Ru | Russian |
| Swa | Swahili |
| Ta | Tamil |
| Uk | Ukrainian |
| Ur | Urdu |
| Wol | Wolof |

Table 12: Languages and their codes

| Group | Task | Train set | Train set size (combined) | Dev set size |
|-------|------|-----------|---------------------------|--------------|
| Indo-Aryan | NER | {En,Hi,Bn,Ur} | 55026 | 13003 |
| Germanic | POS | {En,Is,De} | 222792 | 28394 |
| Slavic | POS | {En,Ru,Cs} | 179043 | 23479 |
| African | NER | {En,Amh,Swa,Wol} | 19788 | 4073 |

Table 13: Training & Dev size (# Sentences)

| Group | Task | Train set | LA set | Test set |
|-------|------|-----------|--------|----------|
| Indo-Aryan | NER | En,Hi,Bn,Ur | {En,Hi,Bn,Ur} | {As,Bh} |
| Germanic | POS | {En,Is,De} | {En,Is,De} | {Fo,Got,Gsw} |
| Slavic | POS | {En,Ru,Cs} | {En,Ru,Cs} | Qpm,Hsb,Orv,Cu |
| African | NER | {En,Amh,Swa,Wol} | {En,Amh,Swa,Wol} | {Hau,Ibo,Kin,Lug,Luo,Pcm} |

Table 15: Language groups and their corresponding train, LA and test sets

| Test Language | Script | Size | Overlap(%) |
|---|---|---|---|
| Fo | Latin | 1208 | 61.7 |
| Got | Latin | 1031 | 21.6 |
| Gsw | Latin | 100 | 54.3 |
| Qpm | Cyrillic | 635 | 42.9 |
| Hsb | Latin | 626 | 43.1 |
| Orv | Cyrilic | 4204 | 45 |
| Cu | Cyrillic | 1141 | 19.2 |
| As | Bengali | 100 | 44.1 |
| Bh | Devanagri | 102 | 64 |
| Hau | Latin | 570 | 37.1 |
| Ibo | Latin | 642 | 32 |
| Kin | Latin | 611 | 32.6 |
| Lug | Latin | 419 | 28.3 |
| Luo | Latin | 189 | 35.8 |
| Pcm | Latin | 600 | 82.3 |

Table 14: Test languages with their scripts, size (# Sentences) and token-level overlap (in % IOU) with source training data

| Group | Task | Tag Set | Test Lang Set |
|---|---|---|---|
| Indo-Aryan | NER | {PER,LOC,ORG} | {As,Bh} |
| Germanic | POS | TagPOS | {Fo,Got,Gsw} |
| Slavic | POS | TagPOS | {Qpm,Hsb,Orv,Cu} |
| African | NER | {PER,LOC,ORG,DATE} | {Hau,Ibo,Kin,Lug,Luo,Pcm} |

Table 16: Language groups and their corresponding tag sets. TagPOS = {PART,CONJ,ADJ,ADP,ADV,VERB,DET,INTJ, NOUN, PRON, PROPN, NUM, PUNCT, AUX, SYM, X}

# E    Qualitative Analysis

We also did some qualitative analysis to understand from where does the actual gain of ZGUL come from. To do this, for both POS and NER tasks, we examined label-wise performance of our model, vis-a-vis the baselines. For POS, we see that ZUGL does quite well on 'NOUN' which has a huge support, resulting in overall better performance for ZUGL. ZUGL does somewhat worse on labels such as 'CONJ'. Similarly, for NER, ZUGL is able to do well on labels such as 'LOC' and 'PER', resulting in overall improved performance over its competitors. We refer to the Appendix E.1 for further details. Carefully examining the reasons behind improved performance on certain labels (while worse performance on others) is a direction for future work.

## E.1    Examples

| Model | | | | | Labels | | | | | |
|---|---|---|---|---|---|---|---|---|---|---|
| Sentence | Daas | Buech | laufft | besser | als | jede | vo | sine | Krimi | . |
| Gold Labels | DET | NOUN | VERB | ADV | CONJ | PRON | ADP | DET | NOUN | PUNCT |
| ZGUL | DET | NOUN | VERB | ADV | CONJ | PRON | ADP | DET | NOUN | PUNCT |
| CPG | DET | PROPN | VERB | ADV | ADP | DET | ADP | DET | NOUN | PUNCT |

Table 17: Example from the gsw language

| Model | Labels | | | | | | | |
|---|---|---|---|---|---|---|---|---|
| Sentence | весь | днь | милует | и | в | заимъ | даеть | праведныи |
| Gold Labels | DET | NOUN | VERB | CONJ | ADP | NOUN | VERB | ADJ |
| ZGUL | DET | PUNCT | VERB | PART | ADP | NOUN | VERB | ADJ |
| CPG | DET | PUNCT | VERB | CONJ | ADP | NOUN | VERB | ADJ |

Table 15: Example from the orv language

Figure 9: Orv language

| Model | Labels | | | | | | | | | |
|---|---|---|---|---|---|---|---|---|---|---|
| Sentence | Jami'an | tsaron | Lebanon | ... | Jakadancin | Amurka | da | ke | birnin | Beirut |
| Gold Lables | O | O | LOC | ... | O | LOC | O | O | O | LOC |
| ZGUL | O | O | LOC | ... | O | LOC | O | O | O | LOC |
| SFT-M | O | O | LOC | ... | PER | PER | O | O | O | LOC |

Table 18: Example from the hau langauge

| Model | Labels | | | | | | | | | | |
|---|---|---|---|---|---|---|---|---|---|---|---|
| Sentence | Doho | ... | pachoka | David | Maraga | chiwo | ... | jii | 11 | matiyo | ... |
| Gold Labels | O | ... | O | PER | PER | O | ... | O | O | O | ... |
| ZGUL | O | ... | O | PER | PER | O | ... | O | DATE | DATE | ... |
| SFT-M | O | ... | O | PER | PER | O | ... | O | O | O | ... |

Table 19: Example from the luo language

# F   Class-wise F1 scores

Table 20 and Table 21 show the classwise F1 scores for each of the tasks. The scores are averaged over all languages in each task.

We have used the seqeval[10] framework for evaluating all the models, which is consistent with the previous works and used by XTREME[11]. Seqeval removes the 'B' and 'I' prefixes of the labels, hence the Table 21 has only 4 classes (E.g. 'B-PER' and 'I-PER' are mapped to same label 'PER')

[10]https://github.com/chakki-works/seqeval
[11]https://github.com/google-research/xtreme

| Class | ZGUL | CPG | SFT-M | Support |
|---|---|---|---|---|
| PART | **29.2** | 26.3 | 25.7 | 831 |
| CONJ | 63.8 | **64.6** | 64.1 | 9000 |
| ADJ | **44.3** | 44.1 | 43.6 | 8768 |
| ADP | **78.8** | 77.2 | 75.1 | 9960 |
| ADV | 27.7 | 27.6 | **27.8** | 5830 |
| VERB | **51.2** | 51.1 | 50.4 | 14385 |
| DET | 40.4 | **40.8** | 37.2 | 3027 |
| INTJ | 10.4 | 3.3 | **13.8** | 211 |
| NOUN | **58.9** | 57.3 | 58 | 18957 |
| PRON | **41.2** | 41 | 40.4 | 7068 |
| PROPN | **54.8** | 52.8 | 53.2 | 4506 |
| NUM | **58.4** | 57.2 | 57.3 | 1508 |
| PUNCT | **62.3** | 61 | 62.1 | 7670 |
| AUX | **42** | 41.2 | 40.3 | 2800 |
| SYM | 11.8 | **13.3** | 12.2 | 33 |
| X | **3.8** | **3.8** | 2.4 | 228 |
| Micro-F1 | **55.9** | 54.4 | 54.1 | 94782 |

Table 20: Classwise F1-scores for POS task. Averaged over all languages

| Class | ZGUL | SFT-M | CPG | Support |
|---|---|---|---|---|
| DATE | 21.2 | **22.1** | 21.9 | 623 |
| LOC | **55.2** | 53.4 | 50.3 | 1643 |
| ORG | **53** | **53** | 48.4 | 1034 |
| PER | **66.7** | 64.2 | 64.1 | 1483 |
| Micro-F1 | **57.1** | 55.6 | 53.4 | 4783 |

Table 21: Classwise F1-scores for NER task. Averaged over all languages

# G  Standard Deviation Details

| | Germanic | | | | Slavic | | | | |
|---|---|---|---|---|---|---|---|---|---|
| | Fo | Got | Gsw | **Avg** | Qpm | Hsb | Orv | Cu | **Avg** |
| MADX$^{multi}$-EMEA | 0.5 | 0.8 | 0.4 | 0.5 | 0.3 | 0.2 | 0.2 | 0.7 | 0.4 |
| SFT-M | 0.3 | 1.2 | 0.6 | 0.7 | 0.4 | 0.3 | 0.2 | 0.4 | 0.1 |
| CPG | 0.1 | 0.5 | 0.2 | 0.3 | 0.3 | 0.2 | 0.0 | 0.2 | 0.0 |
| ZGUL | 0.0 | 0.5 | 0.4 | 0.1 | 0.5 | 0.5 | 0.1 | 0.3 | 0.1 |

Table 22: F1 Std. Dev. (rounded to 1 decimal) of POS Tagging for Germanic and Slavic language groups. Note:-Avg column denotes std. dev. of average F1 and not the average of std dev.

|  | African | | | | | | | | Indo-Aryan | | |
|---|---|---|---|---|---|---|---|---|---|---|---|
|  | Hau | Ibo | Kin | Lug | Luo | Pcm | **Avg** | | As | Bh | **Avg** |
| MADX$^{multi}$-EMEA | 1.4 | 1.2 | 0.9 | 0.4 | 0.5 | 1.4 | 1.1 | | 1.2 | 1.3 | 0.5 |
| SFT-M | 1.3 | 0.8 | 0.7 | 0.7 | 0.6 | 0.5 | 0.7 | | 1.4 | 1 | 0.9 |
| CPG | 1.3 | 0.9 | 1.4 | 0.9 | 0.9 | 0.8 | 1 | | 1.1 | 1.5 | 0.8 |
| ZGUL | 1.1 | 0.9 | 0.7 | 0.4 | 0.5 | 0.3 | 0.4 | | 0.7 | 1.1 | 0.5 |

Table 23: F1 Std. Dev. (rounded to 1 decimal) of NER for African and Indo-Aryan language groups. Note:- Avg column denotes std. dev. of average F1 and not the average of std dev.

## H Language-wise Few Shot Performance

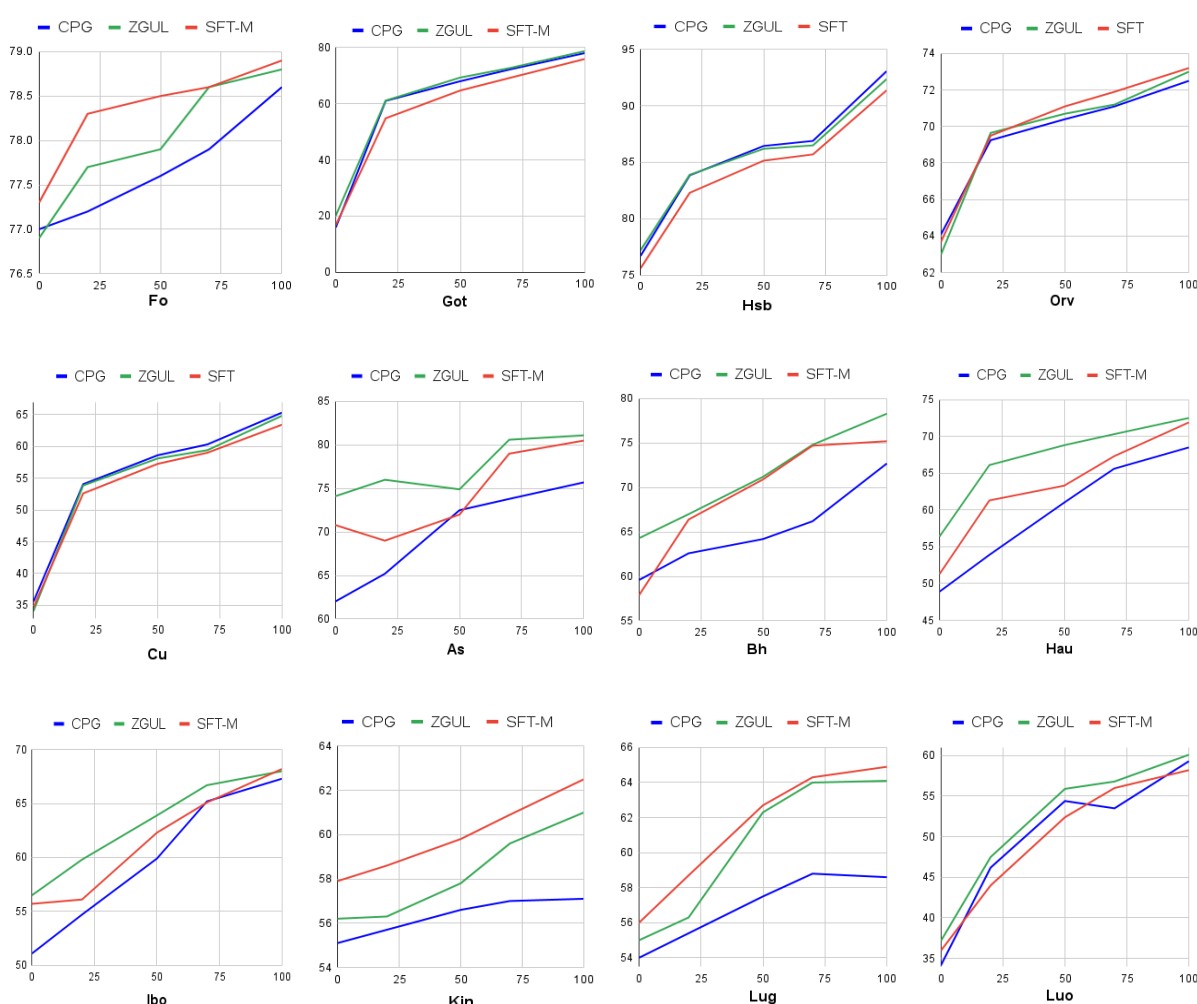

Figure 10: Language-wise few-shot performance results