# OpenReview forum: "ZGUL: Zero-shot Generalization to Unseen Languages using Multi-source Ensembling of Language Adapters"
_EMNLP/2023/Conference — EMNLP 2023 Main_

### Official Review · Reviewer_m43w · 2023-08-03

**Soundness:** 4

**Excitement:**

3: Ambivalent: It has merits (e.g., it reports state-of-the-art results, the idea is nice), but there are key weaknesses (e.g., it describes incremental work), and it can significantly benefit from another round of revision. However, I won't object to accepting it if my co-reviewers champion it.

**Paper Topic And Main Contributions:**

This paper proposes a method ZGUL for improving zero-shot transfer between languages using ensembles of adaptors.
The method is a combination of several methods:
1. Train-time ensembling through token-based attention
2. Train-time ensembling through language vectors
3. Test-time ensembling through EMEA

**Reasons To Accept:**

Overall, I found this paper solid. It is well-aware of previous work and combines it together in a reasonable way.

The experimentation is also relatively thorough. I appreciate all of it, but particularly appreciate section 4.4, where unlabeled target language data is incorporated into training, which is a practical setting.

I also appreciate that the code will be released, which will greatly improve reproducibility.

**Reasons To Reject:**

The main disadvantage of this work is that it is somewhat incremental methodologically. It is mainly based on a combination of several existing methods. Overall complexity can be warranted when it improves results (as it seems to do in this paper), but it also can be detrimental to further progress compared to simpler methods.

**Reproducibility:**

5: Could easily reproduce the results.

**Reviewer Confidence:**

4: Quite sure. I tried to check the important points carefully. It's unlikely, though conceivable, that I missed something that should affect my ratings.

---

> ### Author Rebuttal · Authors · 2023-08-29
>
> Dear reviewer, \
> We are grateful to you for your valuable time to provide us with your feedback. Also, many thanks for appreciating our work. \
> We are happy to address your concerns as follows -
>
> **Lack of Novelty** -
> * _AdapterFusion_ - To our knowledge, we are the first to apply AdapterFusion (Pfeiffer et al., 2021a) in context of Language Adapters (LAs) and find that this standalone technique in itself is not sufficient (as can be verified from ablation results) and hence, explored other ideas such as use of language vectors to aid the model.
> * _Lang2Vec_ Attention - To our knowledge, we are the first to introduce the concept of global attention over LAs as a function of  language vectors (Lang2Vec). In contrast, the CPG method used these language vectors to generate the entire parameters of a (single) shared Adapter. So, we claim this to be our novel component in the model.
> * _EM_ at inference - Wang et al., 2021 proposed EMEA - a pure inference-time algorithm. We modified it with the following amendments for our setting - (1) We made each token-level attention weight in each layer trainable, while the original EMEA had tied it layer-wise. This gives the EM method more degree of freedom in our framework compared to EMEA. (2) We have 2 attention networks, each initialized with its respective trained attention scores, while the original technique had only a single network, initialized naively with uniform weights. \
> To show why amendment (1) is useful (though marginally), we tie the token-level weights layer-wise (as in EMEA) and compare with our EM approach as follows -
> | Model | Hau | Ibo | Kin | Lug | Luo | Pcm | As | Bh | Fo | Got | Gsw | Qpm | Hsb | Orv | Cu | Average |
> | -------- | ------- | ------- | ------- | ------- | ------- | ------- | ------- | ------- | ------- | ------- | ------- | ------- | ------- | ------- | ------- | ------- |
> | Layer-wise Tied as in original EMEA | 53.7 | 57.5 | 55.6 | 54.5 | 40.5 | 66 | 74.1 | 65.4 | 76.6 | 19.2 | 62.7 |50.1 |76.9 | 62.9 | 34.2 | 56.7 |
> | ZGUL | 54.1 | 57.4 | 56.6 | 54.6 | 40.9 | 66.7 | 74.4 | 65.4 | 77.1 | 20.8 | 65.6 | 50.7 | 77.4 | 63.6 | 34.9 | 57.3 |
>
> Also, amendment (2) naturally follows from the components of our architecture itself.
>
> Moreover, our understanding of local and global attention mechanisms for LA ensembling is also novel as far as cross-lingual transfer literature is concerned.
>
> Overall, each of the previous works have been adapted with suitable modifications for our setting and then combined novelly to yield a final strong working solution.
>
> **Complexity of the proposed method**: \
> As far as overall complexity of ZGUL is concerned, we can infer from Appendix A.1 that ZGUL has around 23% and 16 % of the trainable parameters of SFT and CPG, respectively. This parameter efficiency is decent, given the degree of gains achieved by ZGUL in various settings compared to these methods.
>
> As promised, we release our code & models in order to enable future researchers to build further on top of our proposed methodology. We have mentioned the scope for future work as well as current limitations of our work (Sections 5 and 6, respectively) that are open challenges to be addressed by the community. We also believe that some complexity is acceptable as long as it significantly improves the results, which we show it does, in this case.

---

### Official Review · Reviewer_RmJP · 2023-08-05

**Soundness:** 4

**Excitement:**

4: Strong: This paper deepens the understanding of some phenomenon or lowers the barriers to an existing research direction.

**Missing References:**

- Line 206: missing citation Bahdanau attention
- Line 285 - CItation required for UDPOS
```
@article{nivre2018universal,
  title={Universal Dependencies 2.2},
  author={Nivre, Joakim and Abrams, Mitchell and Agi{\'c}, {\v{Z}}eljko and Ahrenberg, Lars and Antonsen, Lene and Aranzabe, Maria Jesus and Arutie, Gashaw and Asahara, Masayuki and Ateyah, Luma and Attia, Mohammed and others},
  year={2018}
}
```

- Line 287 - Citation required for PAN-X
```
@inproceedings{Pan2017,
author = {Pan, Xiaoman and Zhang, Boliang and May, Jonathan and Nothman, Joel and Knight, Kevin and Ji, Heng},
booktitle = {Proceedings of ACL 2017},
pages = {1946--1958},
title = {{Cross-lingual name tagging and linking for 282 languages}},
year = {2017}
}
```
- Line 202, 457 - missing citation mBERT
- Line 442, 456 - Citation missing for XLMR


**Paper Topic And Main Contributions:**

The goal of the paper is to propose a new method to extend already trained multilingual models to languages unseen during the pretraining phase. In particular they make use of language adaptors from a set of related languages to extend the capabilities to newer languages. It is important to note that the scripts for unseen languages are to be already present in the existing model. The methodology is described as follows, (i) language adapters from related languages fused together with adapter fusion, (ii) additional meta data is provided in the form of syntactic and phonological properties, and (iii) entropy maximization of adapter weights for inference. The experiments are run on 15 languages across 4 language families and 2 tasks. The proposed methodology leads to improved gains across most of the languages.

**Reasons To Accept:**

- The paper proposes a new approach to extend multilingual language models to newer unseen languages.
- Relatively inexpensive approach given that the script of the language is already represented in the existing multilingual model
- The paper performs complete ablations of all the proposed components and show the utility of the proposed approach
- They perform qualitative analysis of the outputs and also show the interpretable nature of the features used for training. Solidifying the results obtained.
- Authors release the code and the trained models


**Reasons To Reject:**

- This is not a criticism of the main work, but the limitations section is non-informing. The section needs to provide crucial cases where the approach fails or show potential failure conditions. And possibly shed some light on future directions. For example, (i) the approach not tested or may not be useful for scripts unseen is a possible limitation of the work (ii) the approach not tested for language generation, etc.

**Reproducibility:**

5: Could easily reproduce the results.

**Reviewer Confidence:**

3: Pretty sure, but there's a chance I missed something. Although I have a good feel for this area in general, I did not carefully check the paper's details, e.g., the math, experimental design, or novelty.

**Typos Grammar Style And Presentation Improvements:**

- I would suggest improving figure 1, by increasing the font size of the text, arrows, providing a legend with the acronyms and operations symbols, export higher quality figures. Preferably pdf
- Line 082 - first usage of acronym. Use the full form here than line 144
- Line 251 - tained -> trained
- Line 256 - Algo 11 -> Algorithm 1 (use \Cref)
- Line 290 - Remove additional full stop
- Line 292 - Add full stop. “adapters available.”
- Table 1 - Does “Is” imply Icelandic?
- Consider changing language codes to “each language with a BCP 47 tag sequence using a three-letter ISO 639-3 code as the base subtag and complement with ISO 15924 script subtags” for example - Hindi - “hin_Deva”. As done in https://arxiv.org/abs/2207.04672
- Line 319 - xtreme -> XTREME
- Line 336 - move citation to the comma, for better readability
- Line 339 - What is POR?
- Line 339 - Mention Appendix section/number
- Line 351-353 - ill formed sentence
- Table 2 & 3 - have the same degree of precision for all entries. Example - Use 70.0 instead of 70
- Table 2 & 3 - write explanatory captions with brief descriptions of the 3 sections of the table
- Line 371 - Remove additional full stop
- Line 371 - Table -> Figure
- Line 401 - indic -> Indic
- Line 457 - m-BERT -> mBERT, maintain consistency
- Section 4.6 - multiple instances of ZUGL -> ZGUL

---

> ### Author Rebuttal · Authors · 2023-08-29
>
> Dear reviewer, \
> We express our gratitude for your valuable time to provide us with your feedback. Also, many thanks for citing these many reasons for our work to be published.
>
> We like to address your concerns as follows - \
> **Rewriting Limitations section:** \
> As far as the limitations section is concerned, yes we would move some items from future work to there in the camera-ready version. The final text for this section would be as follows -  \
> “Our method incurs high inference-time overhead for each forward pass owing to Adapters being inserted in each layer. Further, the entropy minimization typically needs 5 or 10 forward passes for effective performance, which leads to further multiplying factor with each forward pass time. These trade-offs between performance and cost are inherited from Wang et al, 2021 itself. Language Adapters have been trained on the Wikipedia dump of source/target languages. This might potentially impose restrictions to extending our technique's efficacy to domain-specific tasks not having sufficient publicly available data in that language and domain as to train a strong Adapter (E.g. Medical domain for African languages). Presently, our technique cannot be tested directly on unseen scripts because our tokenization/embedding layer is same as that of mBERT and may become a bottleneck for Adapters to directly perform well. Our approach is not currently tested on deep semantic tasks and generation-based tasks owing to the lack of suitable large-scale datasets for evaluation.”
>
> **Missing citations and typos**: \
> We really appreciate you taking your time to figure out missing citations and other typos. However, it seems that these typos are not consistent w.r.t. our latest version, but are valid for our previously submitted version. It appears that you mistakenly downloaded that PDF, instead of our main submission while checking for these typos. \
> We humbly point you to our main submission PDF (link - https://openreview.net/pdf?id=aIp5EZeO3f ). Kindly note that most of these typos have already been fixed in the main version. A few remaining ones would be incorporated in the camera-ready version. \
> Moreover, an important section (Section 4.4) is completely new, and was added on the basis of the comments made by the previous set of reviewers. You might enjoy reading this as well.
>
> Thank you once again for your meticulous review!

---

### Official Review · Reviewer_itj7 · 2023-08-05

**Soundness:** 4

**Excitement:**

3: Ambivalent: It has merits (e.g., it reports state-of-the-art results, the idea is nice), but there are key weaknesses (e.g., it describes incremental work), and it can significantly benefit from another round of revision. However, I won't object to accepting it if my co-reviewers champion it.

**Paper Topic And Main Contributions:**

This research piece deals with the problem of the support of language models for unseen low-resource languages, i.e. languages that are not part of the training data, nor have a corpus available.
To that end the authors propose an ensemble of language-specific adapters that have some degree of relatedness with the unseen languages. During train time, the model learns to combine language adapters of previously seen languages, and uses a language specific attention to bring additional information.
At test time, the model uses Entropy-Minimization from Wang et al., 2021).
The model is evaluated in terms of two NLP tasks, part-of-speech (POS) tagging and named
entity recognition (NER) and compared to baselines as mBERT and other variations of MADX with different training strategies, and changing the amount of languages as part of the finetuning.
To finish, the work include a few ablation studies, as an analysis interpretability of the attention scores and the results if there is a minimal amount of data from the low resource language to use.

**Questions For The Authors:**

What’s the training objective of the train-time Ensemble? It seems it’s not clarified on section 3.1

**Reasons To Accept:**

The work address a realistic scenario (zero shot on extremely low resource languages)

The approach is reasonable, it has been built on top of established and proven techniques such as AdapterFusion.

Results, specifically on Tables 2 and 3 show an improvement in performance compared to the baselines. Also, proves that all the components are needed to clearly surpass all the baselines.

**Reasons To Reject:**

The proposed approach lacks enough novelty. It proposes an ensemble of adapters, fused with an already known technique (AdapterFusion, Pfeiffer et al., 2021a), global attention brought by Language-vector-based Attention (Lang2Vec, Littell et al., 2017) and an already established algorithm (Entropy-Minimization from Wang et al., 2021).

The approach is weak in motivation. The amendment of the language-vector attention is reasonable but is nowhere mentioned as a previous problem that needed to be solved.

The division of the languages on non existing "African" or "Indian" is wrong (see comments on Table 1 below). This division is technically wrong, and unfortunately, prevents the paper to be accepted as it is.

**Reproducibility:**

5: Could easily reproduce the results.

**Reviewer Confidence:**

4: Quite sure. I tried to check the important points carefully. It's unlikely, though conceivable, that I missed something that should affect my ratings.

**Typos Grammar Style And Presentation Improvements:**

Line 13: This sentence is confusing. Authors are dealing with the problem of zero-shot cross-lingual transfer using adapters. Given the lack of corpora from low resource languages, authors *propose* to tackle the problem by leveraging multiple linguistically related languages. From the last statement is not possible to conclude that there leverage multiple LAs is *needed*. It is proposed, and it works as shown in the rest of the paper. Finally, authors never address the optimality of their cross-lingual transfer, only show that the approach provides better performance compared to previous ones.

Line 39: The training techniques are not disjoint. SFT is a training objective “fine-tune PLMs on task-specific training data in source languages”. Language adapters also fine tune on “ task-specific training data in source languages” but only a new portion of the model is updated (the adapters) while keeping the rest of the model frozen. Authors might like to change (1) definition to full-model fine-tuning instead.

Line 50 This sentence is not clear. In both cases the models are trained with data from a target language. The only difference is that on the latter, the only part of the model that is trained is the adapter.

Line 63 See comment on line 13. Authors propose to use multiple LAs at training time to improve cross-lingual transfer, but nothing implies that people *should* use the approach.

Table 1 The “language families” presented in this table are not completely right. There is not an “African” family. There is an Afro-Asiatic language family, but not even this category clusters all the languages in this list. Languages from this list belong to different language families: Amharic is semitic (Afro-Asiatic), Wolof is Niger-Congo and Swahili belongs to the Bantu family.
Hauza is a Chadic language, belonging to Afro-Asiatic family as well. Igbo is a Niger-Congo language, as Wolof. Kinyarwanda and Luganda are Bantu as Swahili, Luo is a Nilo-Saharan language, from a language family not included in the training.
There is no “Indic” family. All languages listed in this table below to different branches of the Indo-Aryan family.
Finally, Swiss German is a dialect from German, as well as Nigerian Pidgin, which could be considered an English dialect.

Line 342 MADX-{X}, seems to be based on the MAD-X approach from Pfeiffer et al., (2020b) but it's never stated in the paper. Please clarify.

---

> ### Author Rebuttal · Authors · 2023-08-28
>
> Hello reviewer, \
> We would like to duly address your concerns as follows-
> 1. **Lack of Novelty** -
> * _AdapterFusion_ - To our knowledge, we are the first to apply AdapterFusion (Pfeiffer et al., 2021a) in context of Language Adapters (LAs) and find that this standalone technique in itself is not sufficient (as can be verified from ablation results) and hence, explored other ideas such as use of language vectors to aid the model.
>
> * _Lang2Vec_ Attention - Littell et al., 2017 just provided the typological language vectors for 1000+ languages with no insight whatsoever on how to use them. We are the first, to our knowledge, to introduce the concept of global attention over LAs as a function of those language vectors. So, we claim this to be our novel component in the model.
>
> * _EM_ at inference - Wang et al., 2021 proposed EMEA - a pure inference-time algorithm. We modified it with the following amendments for our setting - (1) We made each token-level attention weight in each layer trainable, while the original EMEA had tied it layer-wise. This gives the EM method more degree of freedom in our framework compared to EMEA. (2) We have 2 attention networks, each initialized with its respective trained attention scores, while the original technique had only a single network, initialized naively with uniform weights. \
> To show why amendment (1) is useful (though marginally), we tie the token-level weights layer-wise (as in EMEA) and compare with our EM approach as follows -
> | Model | Hau | Ibo | Kin | Lug | Luo | Pcm | As | Bh | Fo | Got | Gsw | Qpm | Hsb | Orv | Cu | Average |
> | -------- | ------- | ------- | ------- | ------- | ------- | ------- | ------- | ------- | ------- | ------- | ------- | ------- | ------- | ------- | ------- | ------- |
> | Layer-wise Tied as in original EMEA | 53.7 | 57.5 | 55.6 | 54.5 | 40.5 | 66 | 74.1 | 65.4 | 76.6 | 19.2 | 62.7 |50.1 |76.9 | 62.9 | 34.2 | 56.7 |
> | ZGUL | 54.1 | 57.4 | 56.6 | 54.6 | 40.9 | 66.7 | 74.4 | 65.4 | 77.1 | 20.8 | 65.6 | 50.7 | 77.4 | 63.6 | 34.9 | 57.3 |
>
> Also, amendment (2) naturally follows from the components of our architecture itself.
>
> Overall, each of the previous works have been adapted with suitable modifications for our setting and then combined novelly to yield a final strong working solution.
>
>
> 2. **Weak motivation for Lang2Vec-based attention -**
> * Our problem is to improve the current multilingual models for the low-resource languages. The AdapterFusion technique alone doesn't create much impact which motivates us to look for ways to improve the model.
> * Our core hypothesis behind using Lang2Vec-based attention is that an effective combination of source languages’ LAs should be grounded in their linguistic similarity with the target language. We indeed justify our usage of Lang2Vec module in section 4.3 in terms of the correlation scores’ comparison w.r.t. other ensemble-based approaches. This analysis provides evidence for our hypothesis that linguistic similarity should be a crucial factor to be considered for cross-lingual transfer settings.
> * This hypothesis is also reflected in our experimental results (in the form ablations - Tables 2 and 3), where not utilizing language-vector-based attention results in a drop in overall performance.
>
> 3. **Wrong division of African and Indic languages -**
> We clustered the African languages together following the MasakhaNER paper published by Adelani et al 2021. While we will clarify the language families better (as per your comments) in the camera-ready, we note that all of this does not detract from the key point of the paper: that using multiple language data as source, our method obtains better performance on target low-resource language. We test this using high-resource language data (as per line 331), which (we will clarify that) sometimes will include geographically related but linguistically not very related source languages also. \
> The term Indic languages can be used to refer to Indo-Aryan family. See here: https://en.wikipedia.org/wiki/Indic_languages. Nevertheless, we will clarify this also.
>
> * **Questions-**
>
> 1. What’s the training objective of the train-time Ensemble? \
> **Ans**: It’s the word-level cross-entropy loss which is the standard loss function for sequence labeling tasks. We would include this detail in our camera-ready version.
>
> * **Typos Grammar Style And Presentation Improvements:**
>
> 1. "optimal" in Line 13 - We acknowledge that the term “optimal” in line 13 shouldn’t have been used. We will replace it with “more effective” in the camera-ready version.
>
> 2. "From the last statement is not possible to conclude that there leverage multiple LAs is needed"  in Line 13 - This line follows from last two sentences. In line 6 we say that existing works using target LA or single related LA. In line 8 we talk about absence of target LAs for low resource languages. In line 13, we go back to the other issue (single related LA), and posit that for more effective transfer, instead of using one, we should use multiple related LAs. We will clarify this subtle lack of fluency by adding the phrase, “instead of just one source LA”.
>
> 3. Line 39 - We should have concretely mentioned that the acronym SFT represents “fine-tune **all parameters** of a PLM …..”. We are fine changing this acronym to FFT (Full-model Fine-Tuning), if it aids readability.
>
> 4. Line 50 - Yes, your reading is accurate. To better clarify, we will add edit line 53 to say “the latter **also** requires the **same** unlabeled data…”
>
> 5. Line 63 - We do show in our experiments why multiple LAs’ combination is useful. In fact, that is at the core of our work’s thesis. E.g. note that using simply one Adapter, that is most related to a given target language (assuming target LA is not available), has weaker performance (Check out the numbers of MADX-multi-{Rel} vs ZGUL in tables 2 and 3) compared to our method. \
> We further use only a single English Adapter to train our model on multiple languages and use the English adapter at inference. The performance of this approach is significantly lower than ZGUL which utilizes multiple Adapters, while training on the same multilingual data. We present the numbers below:
>
> | Model | Hau | Ibo | Kin | Lug | Luo | Pcm | As | Bh | Fo | Got | Gsw | Qpm | Hsb | Orv | Cu | Average |
> | -------- | ------- | ------- | ------- | ------- | ------- | ------- | ------- | ------- | ------- | ------- | ------- | ------- | ------- | ------- | ------- | ------- |
> | ZGUL w. En LA only | 52.8 | 54.8 | 54.2 | 51.9 | 35.2 | 64.7 | 62 | 60.5	| 74.8 | 14.9 | 60.8 | 48.9	| 74.4 | 62.9 | 34.3 | 53.8 |
> | ZGUL | 54.1 | 57.4 | 56.6 | 54.6 | 40.9 | 66.7 | 74.4 | 65.4 | 77.1 | 20.8 | 65.6 | 50.7 | 77.4 | 63.6 | 34.9 | 57.3 |
>
>
> We clearly observe average gains of using multiple LAs by 3.5 points compared to using only En LA.
>
> 6. Table 1 - We have addressed the "African" and "Indic" issues above. Regarding Swiss German, we don’t understand your concern. Swiss German has been included in the Germanic family only. Also, English is kept a common source language in every family (including "African" that has Nigerian Pidgin as a test language).
>
> 7. Line 342 - Yes, MADX is same as Pfeiffer et al., (2020b). We have cited this method in Line 130 though we didn’t mention the term “MADX” in that paragraph. We would do this edit in our camera-ready version.
>
> Looking at your final scores, we can’t help but wonder that our paper is being judged too harshly and unfairly. While excitement can be more subjective (though scaling NLP technology to low-resource languages is an important area of inquiry today), a soundness score of 1 feels excessive and unreasonable. \
> In particular, almost all issues highlighted by you are minor clarifications; we have answered them in detail, and minor edits in the next version will help clarify them. Finally, we have given a detailed response to your question of novelty, exposing aspects of our contributions that may have been missed in the first version. We hope that in light of this response, you will reassess the value our paper brings to low-resource multilingual NLP and revise your opinion positively. Thank you!

---

### Meta-Review · Area_Chair_jA4P · 2023-09-19

**Recommendation:** 4

**Metareview:**

This paper explores the usage of adapters  to support of language models for unseen low-resource languages during training time. This is done by a proposed architecture called ZGUL that is an ensemble of  ensembles of adapters. This method also claim to use linguistical related source languages to perform this. Overall the method is sound, and tackles a high impact problem. The authors also include a strong ablation study as well as a quantitative analysis. On the reject side, there are some issues that the authors need to tackle (using the correct linguistic family classifications). There are also room for future work as, how to include new scripts, and testing it on other tasks, like generative tasks.

---

### Decision · Program_Chairs · 2023-10-07

**Decision:**

Accept-Main

**Comment:**

This paper explores the usage of adapters  to support of language models for unseen low-resource languages during training time. This is done by a proposed architecture called ZGUL that is an ensemble of  ensembles of adapters. This method also claim to use linguistical related source languages to perform this. Overall the method is sound, and tackles a high impact problem. The authors also include a strong ablation study as well as a quantitative analysis. On the reject side, there are some issues that the authors need to tackle (using the correct linguistic family classifications). There are also room for future work as, how to include new scripts, and testing it on other tasks, like generative tasks.